

# Lygistorrhinidae (Diptera: Bibionomorpha: Sciaroidea) in early Eocene Cambay amber

Frauke Stebner[1,2], Hukam Singh[3], Jes Rust[1] and David A. Grimaldi[4]

[1] Steinmann-Institut, Abteilung Paläontologie, Bonn, Germany
[2] Department of Entomology, Stuttgart State Museum of Natural History, Stuttgart, Germany
[3] Birbal Sahni Institute of Palaeosciences, Lucknow, India
[4] American Museum of Natural History, NY, USA

## ABSTRACT

One new genus and three new species of Lygistorrhinidae in early Eocene Cambay amber from India are described, which significantly increases our knowledge about this group in the Eocene. *Lygistorrhina indica* n. sp. is the oldest fossil known from this extant genus. *Indorrhina sahnii* n. gen. et sp. shows morphological similarities to each of the two extant genera *Lygistorrhina* and *Asiorrhina*. *Palaeognoriste orientale* is the third species known from a group that has only been recorded from Eocene Baltic amber before. The latter finding reveals faunal links between Cambay amber and the probably slightly younger Baltic amber, adding further evidence that faunal exchange between Europe/Asia and India took place before the formation of Cambay amber.

## INTRODUCTION

Lygistorrhinidae is a small dipteran group within the superfamily Sciaroidea (fungus gnats) of the infraorder Bibionomorpha. The monophyly of the latter two groups has been recently confirmed with high support by molecular analysis (*Ševčík et al., 2016*). Within the same analysis relationships of Lygistorrhinidae amongst other Sciaroidea indicates the inclusion of Lygistorrhinidae in the family Keroplatidae. Relationships within Lygistorrhinidae are well explored, with *Parisognoriste* and *Palaeognoriste*, both Eocene in age, being the sister groups to the remaining recent genera, consensus relationships of which are *Asiorrhina* (*Lygistorrhina* (*Labellorhina* + *Blagorrhina* + *Gracillorhina*) + (*Matileola* (*Loyugesa* + *Seguyola*))) (e.g., *Blagoderov, Hippa & Nel, 2010*).

Members of Lygistorrhinidae can be found in tropical to temperate warm forests today (*Grimaldi & Blagoderov, 2001*). They can be easily recognized by the distinct wing venation, with a lack of the stem and sometimes also the base of the fork of M veins, and generally an elongate proboscis, which is presumably for feeding from flowers (*Seguyola* lacks this as a derived condition) (*Grimaldi & Blagoderov, 2001*). Apart from that, however, ecology (e.g., life histories) and distributions of the species remains largely unknown. It has been concluded that rareness of this group in recent collections is due a lack of sampling, which is in turn related to poor knowledge about its biology and distribution

Corresponding author
Frauke Stebner,
frauke.stebner@smns-bw.de,
frauke.stebner@uni-bonn.de

(*Grimaldi & Blagoderov, 2001*). Further, tropical regions like the Neotropics and Asia generally are still poorly collected and probably harbor a huge number of undiscovered species. This is supported by the increase of described genera (number more than tripled) and species (number more than doubled) in the past 15 years, particularly of recent genera from southern Asia (*Grimaldi & Blagoderov, 2001*; *Hippa, Mattsson & Vilkamaa, 2005*; *Blagoderov, Hippa & Ševčík, 2009*) and from fossil taxa (*Blagoderov & Grimaldi, 2004*; *Blagoderov, Hippa & Nel, 2010*; *Grund, 2012*).

To date there are 15 genera, seven fossil and eight extant, and 48 species described (*Pape, Blagoderov & Mostovski, 2011*; *Fungus Gnats Online, 2016*). The fossil record of Lygistorrhinidae dates back to the Cretaceous, with the oldest records from Lebanon (125–129 Ma, *Maksoud et al. (2016)*) (†*Lebanognoriste Blagoderov & Grimaldi, 2004*), Myanmar (98–99 Ma, *Shi et al., 2012*) (†*Archaeognoriste Blagoderov & Grimaldi, 2004*; †*Protognoriste Blagoderov & Grimaldi, 2004*; †*Leptognoriste Blagoderov & Grimaldi, 2004*), Taimyr (Santonian, e.g., *Rasnitsyn et al., 2016*) (†*Plesiognoriste Blagoderov & Grimaldi, 2004*) and Canada (76–80 Ma, e.g., *Borkent, 2000*) (†*Plesiognoriste Blagoderov & Grimaldi, 2004*). There is then a large gap in the fossil record, with the next oldest members known from early Eocene Oise amber (ca. 53 Ma, *Brasero, Nel & Michez, 2009*) (†*Parisognoriste Blagoderov, Hippa & Nel, 2010*), amber from the Baltic Region (35–43 Ma, e.g., *Standke, 2008*) (†*Palaeognoriste Meunier, 1904*) and Dominican amber (15–20 Ma, *Iturralde-Vinent, 2001*) (*Lygistorrhina Skuse, 1890*; see *Grund, 2012*).

Findings of fossil Lygistorrhinidae in Eocene Cambay amber from India now fill a gap in the spatial fossil record of the Palaeogene. Cambay amber occurs in active lignite mines in the state of Gujarat, in western India. The amber-bearing sediments have been dated to 52–55 Ma (*Clementz et al., 2011*; *Garg et al., 2008*; *Punekar & Saraswati, 2010*; *Sahni et al., 2006*). More precisely, vertebrate remains from between the two major amber layers in Vastan mine have been estimated at 54.5 Ma (*Smith et al., 2016*), suggesting an age of ca 54 Ma for the amber.

Thus, Cambay amber is slightly older than Baltic amber and contemporaneous with Oise amber from France and Fushun amber from China (50–53 Ma, *Wang et al., 2014*). Cambay amber has been formed at a climatically pivotal period: namely during the Early Eocene Climatic Optimum (EECO). Further, the time of formation of this amber (or at least its burial) is most likely around the time of collision of the Indian subcontinent with Asia, which, according to most recent stratigraphic results, took place around 59 Ma (*Hu et al., 2015*) and led to the uplift of the Himalaya. In previous studies, investigation of fossils from Cambay amber have proven to be of major significance for the reconstruction of India's past diversity and geological history (e.g., *Rust et al., 2010*; *Stebner et al., 2017*).

Until now, eight fossil Lygistorrhinidae belonging to three species in three different genera have been discovered in Cambay amber. These findings add further evidence that these flies are much more abundant and diverse in past ecosystems than is commonly known, as it has also been suggested for modern faunas. The fossils represent one species that is very similar to fossils of the archaic Baltic amber genus *Palaeognoriste*, as well as two species that belong to more derived, living lineages. Together with known fossil and recent species, and the highly congruent phylogenetic analyses published so far, the finding of

fossils in the early Eocene from India provides interesting information for divergence-time estimations of this family, and eventually for biogeographic studies.

Interestingly, to date no faunal connections between contemporaneous Cambay amber from India and Oise amber from France have been found, though numerous affinities of fossils in Cambay amber to Baltic amber have been recorded (*Rust et al., 2010*; *Engel et al., 2011*; *Engel et al., 2013*; *Grimaldi & Singh, 2012*; *Stebner et al., 2017*). Whether this is a consequence of the relatively nascent state of investigation of the first two amber deposits, compared to the well-known Baltic amber, may eventually be determined by additional future studies on Oise and Cambay amber.

## METHODS AND MATERIALS

The present study is based on eight fossil Lygistorrhinidae in early Eocene Cambay amber from India. The specimens derive from the Tadkeshwar lignite mine (N21°21.400, E073°04.532) in Gujarat, India. Screening of rough amber pieces was done at the Steinmann Institute, Bonn, Germany and the American Museum of Natural History (AMNH), New York, USA. Holotypes are deposited in the Birbal Sahni Institute of Palaeosciences (BSIP), Lucknow India; paratypes retained in the AMNH.

Amber pieces were ground using a Buehler Phoenix Beta grinding machine and Buehler wet-lapidary wheel. For taxonomic identification and investigation a Leica MZ $12_5$ stereoscope and Nikon SMZ1500 were used. Photographs were taken with an AXIO Zoom.V16 stereomicroscope (Carl Zeiss, Jena, Germany) equipped with an AXIOCam HRc digital camera (Zeiss), using the extended-depth-of-focus function, and in New York with a Nikon SMZ1500 stereoscope, digital camera, and Nikon NIS-Elements software.

Photo-plates were edited using Photoshop CS5.1 and Adobe Illustrator CS6.

General morphological terms and abbreviations follow those given in the Manual of Nearctic Diptera (*McAlpine, 1981*) and in *Blagoderov & Grimaldi (2004)*. The spur formula refers to the number of tibial spurs on fore-, mid-, and hind legs. Wing veins are abbreviated as follows: CuA = anterior branch of cubital vein; CuP = posterior branch of cubital vein; h = humeral crossvein; $M_1 - M_4$ = branches of medial vein; $R_1, R_5$ = branches of radial vein; $r - m$ = radial-medial crossvein; Sc = subcostal vein.

The morphological matrix is based on the matrix of *Blagoderov, Hippa & Nel (2010)*. The data matrix was created and edited in Mesquite ver. 2.6 (*Maddison & Maddison, 2017*). Strict consensus cladograms were obtained by using TNT (*Goloboff, Farris & Nixon, 2008*) based on 60 adult morphological characters (after *Blagoderov, Hippa & Nel (2010)*) (matrix shown in Table S1). It was searched by the "implicit enumeration" option with "auto-collapse searches" off. The consensus trees were calculated by "strict (=Nelson)" using all trees and all taxa.

Nomenclatural Acts

The electronic edition of this article conforms to the requirements of the amended International Code of Zoological Nomenclature, and hence the new names contained herein are available under that Code from the electronic edition of this article. This published work and the nomenclatural acts it contains have been registered in ZooBank,

the online registration system for the ICZN. The ZooBank LSIDs (Life Science Identifiers) can be resolved and the associated information viewed through any standard web browser by appending the LSID to the prefix "http://zoobank.org/". The LSID for this publication is: urn:lsid:zoobank.org:pub:DD2F5D2A-C3EE-4F5C-B068-C9E541D9B70C.

## SYSTEMATICS

**Genus *Palaeognoriste* Meunier**

**Type species:** *Palaeognoriste sciariforme Meunier, 1904*: 88, by monotypy. In Baltic amber. *Matile, 1990a*: 360; *Matile, 1990b*: 366, 373–376, 383, 409, 421, 554; *Hoffeins & Hoffeins, 1996*: 311; *Grimaldi & Blagoderov, 2001*: 55; *Hippa, Mattsson & Vilkamaa (2005)*: 5, 11; *Blagoderov, Hippa & Ševčík (2009)*: 32, 33, 35, 37, 45; *Blagoderov, Hippa & Nel (2010)*: 79-81, 83–89.

**Diagnosis** after *Blagoderov, Hippa & Nel (2010)* and *Grimaldi & Blagoderov (2001)*: Head with proboscis no longer than $2.5\times$ head height, palpus one-segmented; wing with complete basal portion of fork of $M_1 - M_2$, vein Rs and crossvein $r - m$ present.

### *Palaeognoriste orientale* Stebner and Grimaldi, new species
LSID urn:lsid:zoobank.org:act:122BEB14-EDDA-45D7-9973-3898B609F39C
Figs. 1A–1E, 2A–2C, 3A–3E, 4A, 4B

**Diagnosis**: Very similar to the two *Palaeognoriste* species in Baltic amber, except that *P. orientale* has a broader, shorter wing; the stem of M and base of M forks are absent (vs. faint); apex of vein CuA is more acutely bent; gonostylus is more curved; apex of the labellar lobes are blunt, not narrowly tapered; clypeus projecting forward, Ω-shaped; antenna is much longer than proboscis (vs. approximately the same length).

**Material**: Holotype, male Tad-887, complete inclusion in early Eocene Cambay amber from India; deposited in the BSIP, Lucknow India. Paratypes, 1 male: Tad-587, 2 females: Tad-37A, Tad-324; deposited in the AMNH.

**Locality**: Cambay Formation (early Eocene), Tadkeshwar lignite mine, Tadkeshwar, Gujarat, India, 21°21.400′N, 73°4.532′E.

**Etymology**: Specific name refers to the origin of the species from the Oriental Region.

**DESCRIPTION:**

**Measurements**: Male. Head height: 0.24 mm; body without head: 1.8 mm; wing measured from humeral vein: 1.06 mm; antenna: 0.8 mm; proboscis: 0.23 mm; palpus: 0.13 mm.

**Head** (Fig. 1E): Globular, occipital setae short, postocellar and frontal setae erect and stronger. Eyes: dichoptic in male and female, medial margins well separated by frons; anterior part extending to base of antenna. Facets round, equal in size (no differentiation), postocular setae present, interocular setae very fine, short, sparse. Number of ocelli three, forming triangle, middle ocellus slightly smaller than lateral ocelli. Palp one-segmented, evenly setose, ca. $0.55–0.6\times$ length of proboscis. Proboscis short, about as long as head high, significantly shorter than profemur; almost bare, with fine hairs on lower surface; apex blunt, not narrowly tapered. Labrum narrowly triangular, without setae, length
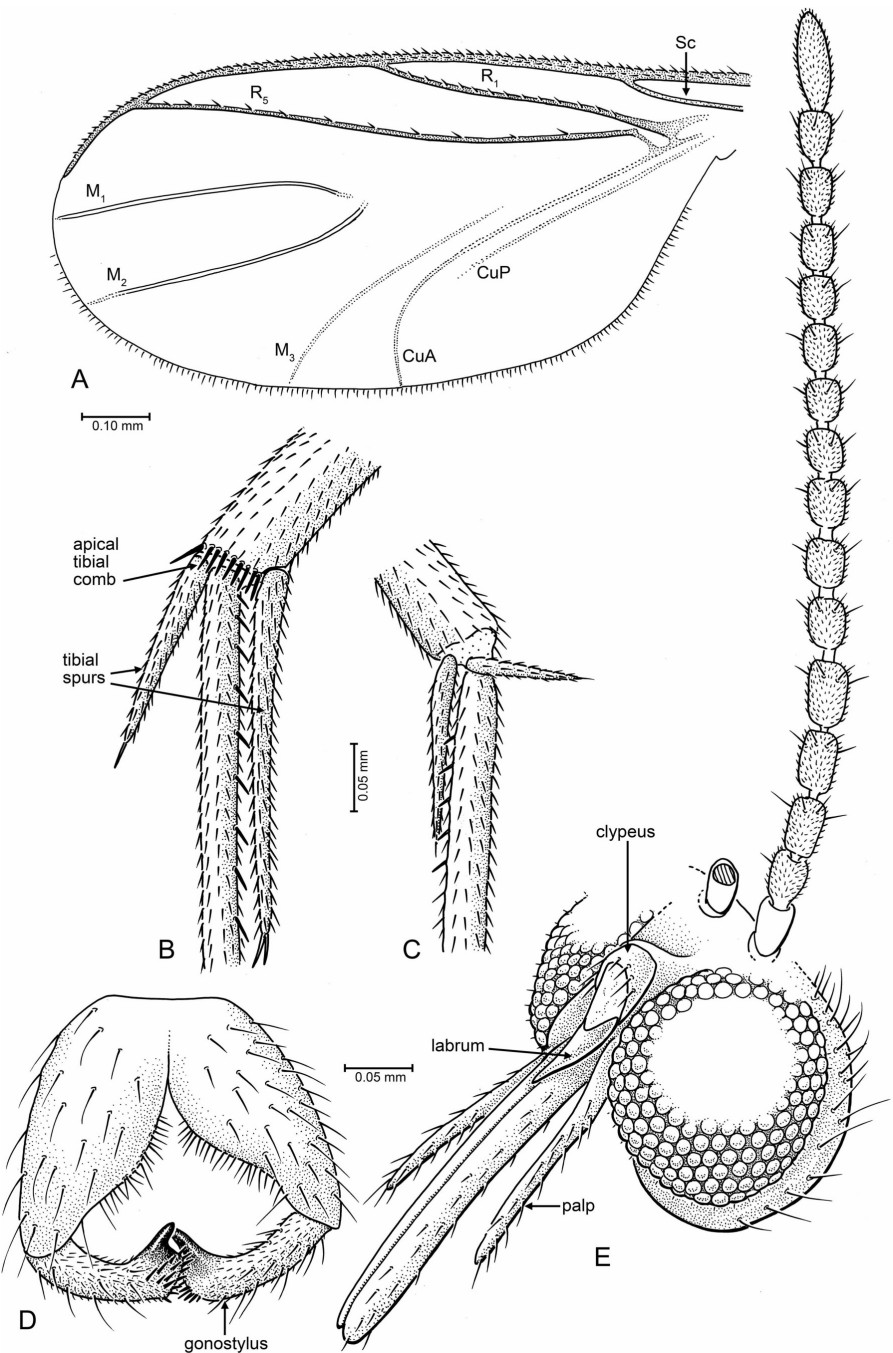

**Figure 1  Drawings of *Palaeognoriste orientale* n. sp., paratype male Tad-587.** (A) Wing. (B, C) Tibial spurs (B, hind; C, mid). (D) Male terminalia, ventral. (E) Head, fronto-lateral.

0.25–0.30× that of proboscis. Antenna with 14 flagellomeres, none laterally flattened. Scape with a slim pedicel/stem and bulbous apical part; pedicel bulbous, 1.8× wider than flagellomeres. Flagellomere 1 subcylindrical; flagellomeres 2–13 cylindrical, about 1.6× longer than broad. Terminal flagellomere elongate, evenly tapered to rounded apex, about 1.5× length of proximal articles and 2.4× longer than broad.

**Thorax**: Scutum moderately convex, not dome-shaped or arched, evenly covered with short setae, longer setae on posterolateral surfaces. Anterior margin of thorax slightly anterior to level of procoxae. Laterotergite only slightly produced, not lobate, with row of 4–5 fine setae on posterior margin. Scutellum with long, stiff setae on posterior margin.

**Legs**: Coxae nearly equal in length (fore coxa nearly equal in length to other two); fore coxa sparsely setose anteriorly, mid coxa sparsely setose anteriorly on apical half, hind coxa bare (holotype) or with very sparse, minute setae (paratype Tad-587). Hind coxa without concavity on lateral surface. Tibiae and tarsi (Figs. 1B, 1C) evenly covered with small setulae, arranged in longitudinal rows. Mid tibia with six thick, erect, evenly-spaced setae. Tibial spurs 1:2:2, length 0.1: (0.2, 0.1): (0.46, 0.26). Fore and hind tibiae slightly expanded towards apex. Short transverse comb of thick setae on apex of hind tibia. All claws with pointed (none with blunt) apex.

**Wing** (Fig. 1A, right wing): length of wing about $2.2\times$ width, apex of wing not broadly rounded. $R_1$ and $R_5$ with row of minute setae, other veins bare. Costa extending beyond $R_5$ tip to 2/3 distance between tips of $R_5$ and $M_1$. Sc ending at C. $R_1$ ending slightly before middle of wing, slightly curved posteriorly. Rs distinct, oblique. Crossvein $r-m$ present. $R_5$ slightly curved in basal third. Stem of M and base of $M_{1+2}$ fork lost. $M_2$ extends more basally than $M_1$. $M_1$ and $M_2$ straight and parallel posteriorly, apices of $M_1$ and $M_2$ on each side of wing apex. $M_{3+4}$ and CuA curved posteriorly, CuA apex in paratype Tad-587 at nearly a right angle to CuA stem, more oblique in holotype (possibly due to fold in the anal area). CuP very close and parallel to stem of CuA.

**Abdomen**: Male terminalia (Fig. 1D): Visible only from posterior in holotype; ventrally in paratype Tad-587. Tergite 9 short and rounded apically, covered with short setae. Gonocoxites wide in basal half, setose, mesal margin with dense small setae. Gonostylus cylindrical, slightly curved; apex beveled, with two large setae—one on each apical edge (these not observed in paratype Tad-587), a patch of thick, short setae between those, and with one subapical lobe-like tooth.

**Additional remarks:**

**Tad-887** (Figs. 2A–2C): Holotype male; preservation: Complete, well displayed, with good preservation, although there is a fracture running straight through the amber matrix. Eyes contorted and sunken by fossilization. Right antenna with flagellomeres $6+7$ and $10+11$ not visibly/distinctly separated, probably due to fossilization processes. Anal lobe of right wing turned down, therefore margin not entirely visible.

**Tad-587** Figs. 3A–3E: Paratype male; preservation: Complete, well displayed, with good preservation, although the fly is encircled by a milky fracture. This, along with compression and some distortion of the thorax, obscures pleural structure. Details of the head, legs, and genitalia are very well preserved. The fly is viewed from its left side only, the right side obscured by an internal translucent fracture plane close to the fly. The area of the head vertex close to the antennae is obscured, so eye emarginations cannot be viewed.

Measurements: Head height: not observable; proboscis length: 0.30 mm; palpus length: 0.20 mm; antenna length: 0.75 mm; body length without head: 1.76 mm; wing length from humeral vein: 1.10 mm.

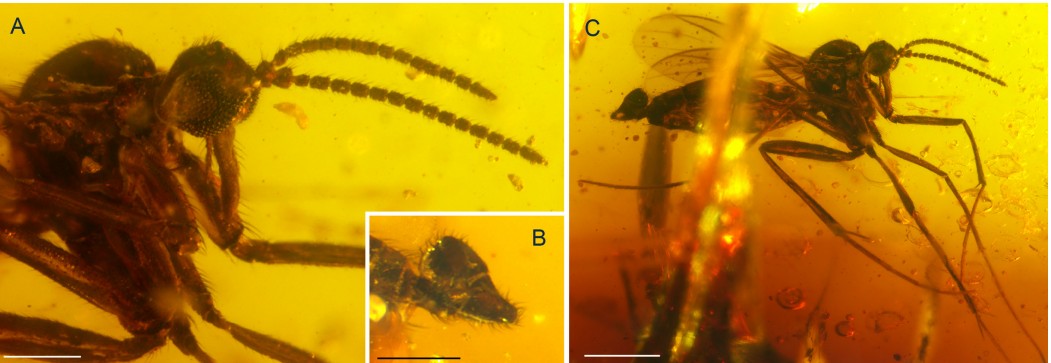

**Figure 2** **Photographs of *Palaeognoriste orientale* n. sp., holotype male Tad-887.** (A) Thorax and head, lateral. (B) Male terminalia. (C) Habitus, lateral view. Scale bars: (A) 0.2 mm, (B). 0.2 mm, (C) 0.5 mm.

**Tad-37A**: Paratype female; preservation: Fair preservation, with legs and antennae well displayed but anterior portion of left wing lost at surface of the amber. Head is partially cleared (dorsum too poorly preserved to observe ocellar area); the fly straddles three fractures, which obscure some structures. The amber required trimming, embedding, and more trimming to obtain reasonable views.

Measurements: Head height: 0.23 mm; proboscis length: 0.31 mm; palpus length (not observable); antenna length: 0.63 mm; body length without head: ca. 1.6 mm; wing length from humeral vein: 1.50 mm.

**Tad-324** (Fig. 4A–4B): Paratype female; preservation: Fair preservation, appendages (including mouthparts and antennae) well exposed, but some legs (especially prolegs) distorted by stretching. Wings intact but partly obscured by fracture over them; terminalia obscured. The amber required considerable trimming, then was embedded prior to more trimming.

Measurements: Head height: 0.29 mm; proboscis length: 0.37 mm; palpus length 0.21 mm; antenna length: 0.43 mm; body length without head: 1.77 mm; wing length from humeral vein: 1.37 mm.

**Genus *Lygistorrhina* Skuse**

**Type species:** Type species: *Lygistorrhina insignis Skuse (1890)*: 600, by monotypy. *Williston (1896)*: 261; *Johannsen (1909)*: 62; *Edwards (1912)*: 203, 204; *Senior-White (1922)*:197; *Edwards (1925)*: 530; *Edwards (1926)*: 245, 246; *Tonnoir (1929)*: 590; *Edwards (1932)*: 139; *Okada (1937)*: 46; *Lane (1946)*: 345; *Shaw & Shaw (1951)*: 16; *Hennig (1954)*:309; *Lane (1958)*: 209, 210; *Hennig (1966)*: 50; *Tuomikoski (1966)*: 254–260; *Thompson (1975)*: 434–444; *Matile (1979(1978)*: 251–255; *Matile (1986)*: 286–288; *Matile (1990a)*: 359–362, 364–370; *Matile (1996)*: 30; *Huerta & Ibañez Bernal (2008)* : 44–51; *Evenhuis (2008)*: 13–19; *Grimaldi & Blagoderov (2001)*: 43–45, 47, 48, 52–54, 56; *Papp (2002)*: 135,138–140; *Papp (2005)*: 151–154; *Hippa, Mattsson & Vilkamaa (2005)*: 2–6, 8, 10, 11, 13, 16, 19; *Blagoderov, Hippa & Ševčík (2009)* : 31-33, 35, 37; *Grund (2012)*: 639–642; *Blagoderov, Papp & Hippa (2013)*: 1, 2, 4–11.

**Diagnosis** after *Grimaldi & Blagoderov (2001)*: Head with proboscis >4× length of head, including palps; wing with vein Sc incomplete or complete, basal portion of fork of $M_1 - M_2$ lost, $CuA_1$ and $CuA_2$ not connected in distal fork. Thorax with laterotergite expanded outward, flap-like, with fringe of long setae on edge.

### *Lygistorrhina indica* Stebner and Grimaldi, new species
LSID urn:lsid:zoobank.org:act:C787AB55-78E4-4E19-940D-29BEAD5A2ABE
**Figs. 5B**, **6C**, **6D**, **7B**, **8A–8D**, **9A**, **9B**

**Diagnosis:** Very similar to recent and Neogene species of *Lygistorrhina* by the very long, slender proboscis and palpi; anterior ocellus small to minute; laterotergite lobate and setose; stem of M and bases of $M_1 - M_2$ fork absent; $R_5$ without setae. Differing from other species of the genus by lack of concavity on lateral surface of metacoxa, and setae on laterotergite not in a row on posterior margin.

**Material**: Holotype, female Tad-442, complete inclusion in early Eocene Cambay amber from India; deposited in the BSIP. Paratypes, 1 female: Tad-492, 1 incomplete specimen of unknown sex: Tad-888, deposited in the AMNH.

**Locality**: Cambay Formation (early Eocene), Tadkeshwar lignite mine, Tadkeshwar, Gujarat, India, 21°21.400′N, 73°4.532′E.

**Etymology**: Specific name refers to the origin of the amber from India.

### DESCRIPTION:

**Measurements**: Measurements: Head height: 0.29 mm; proboscis length: 0.90 mm; palpus length: 0.61 mm; antenna length: 0.58 mm; body length without head: 2.17 mm; wing length from humeral vein: 1.32 mm. Only female known.

**Head** (Fig. 5B): Subspherical, occiput and vertex short setose. Eyes with facets round and equal in size, ommatidia very densely set. Postocular setae not present or not visible, interocular setae short, sparse, very fine. Eyes emarginate just lateral to antennal base; emarginate area bare of ca. one row of facets. Number of ocelli three, nearly in a transverse line, middle ocellus significantly smaller than lateral ocelli. Frons and face broad; extensive membranous area ventral and lateral to antennal bases, with small sclerotized condyle. Clypeus slightly crescentic, with marginal setae. Palpi one-segmented, long, 0.7× the length of proboscis, tapered apicad to fine tip, with a single row of setae. Proboscis evenly curved (of medium length), (very long in Tad-442, Tad-492) about 2.2× (3.3×) as long as head high, with fine short setae. Oral margin slightly protruding; deeply incised, with clypeus nestled fully within. Labrum long triangular, with rounded apex, without setae. Antenna with 14 flagellomeres. Scape bulbous; pedicel subcylindrical, broadened apically, 1.8× wider than flagellomeres. Flagellomeres 1–13 cylindrical, about 1.2× longer than broad. Terminal flagellomere evenly tapering to rounded apex, only slightly longer than previous ones and 1.5× longer than broad.

**Thorax** (Fig. 6D): Scutum dome-shaped, strongly arched, entirely covered with setae of medium length; acrostichals long, longer setae on posterolateral surface of scutum. Scutellum with a row of long setae on posterior margin. Thorax in lateral view short

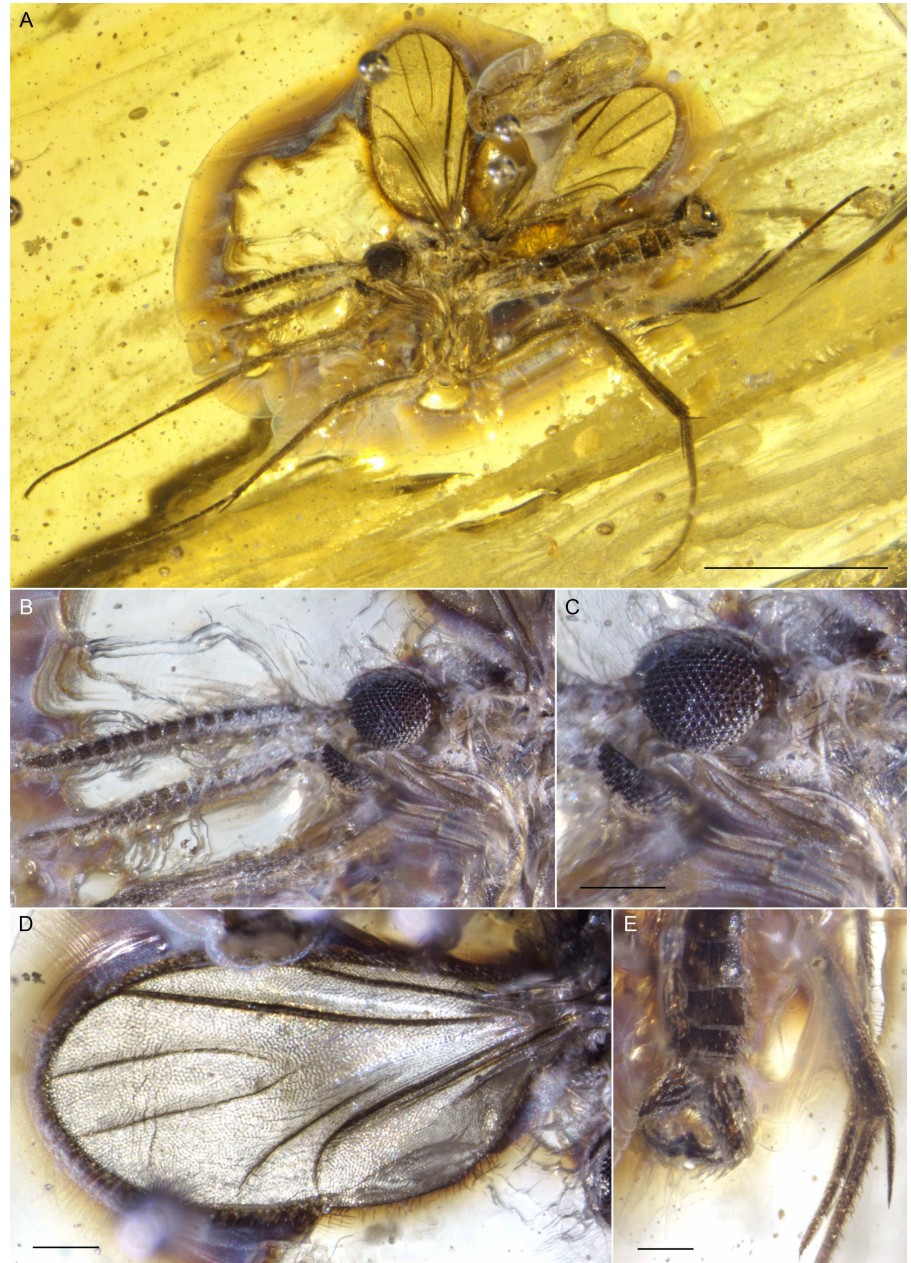

**Figure 3** **Photographs of *Palaeognoriste orientale* n. sp., paratype male Tad-587.** (A) Habitus. (B) Head. (C) Eyes. (D) Wing. (E) Male terminalia. Scale bars: (A) 1 mm, (C) 0.1 mm, (D) 0.1 mm, (E). 0.1 mm.

(anterior to posterior margins), tall/deep (dorsal to ventral margins). Laterotergite well produced posteriad into a lobe, with long setae over broad surface (not in marginal row). **Legs**: Coxae virtually equal in length. Fore coxae setose anteriorly, mid and hind coxae with sparse setae laterally. Fore tibia evenly covered with short setae, with one tibial spur, not expanded towards apex, approximately equal in length to fore femur. Tibiae and tarsi with microtrichia in longitudinal rows. Tibial spurs 1 : 2 : 2; anterior midtibial spur 0.3× length

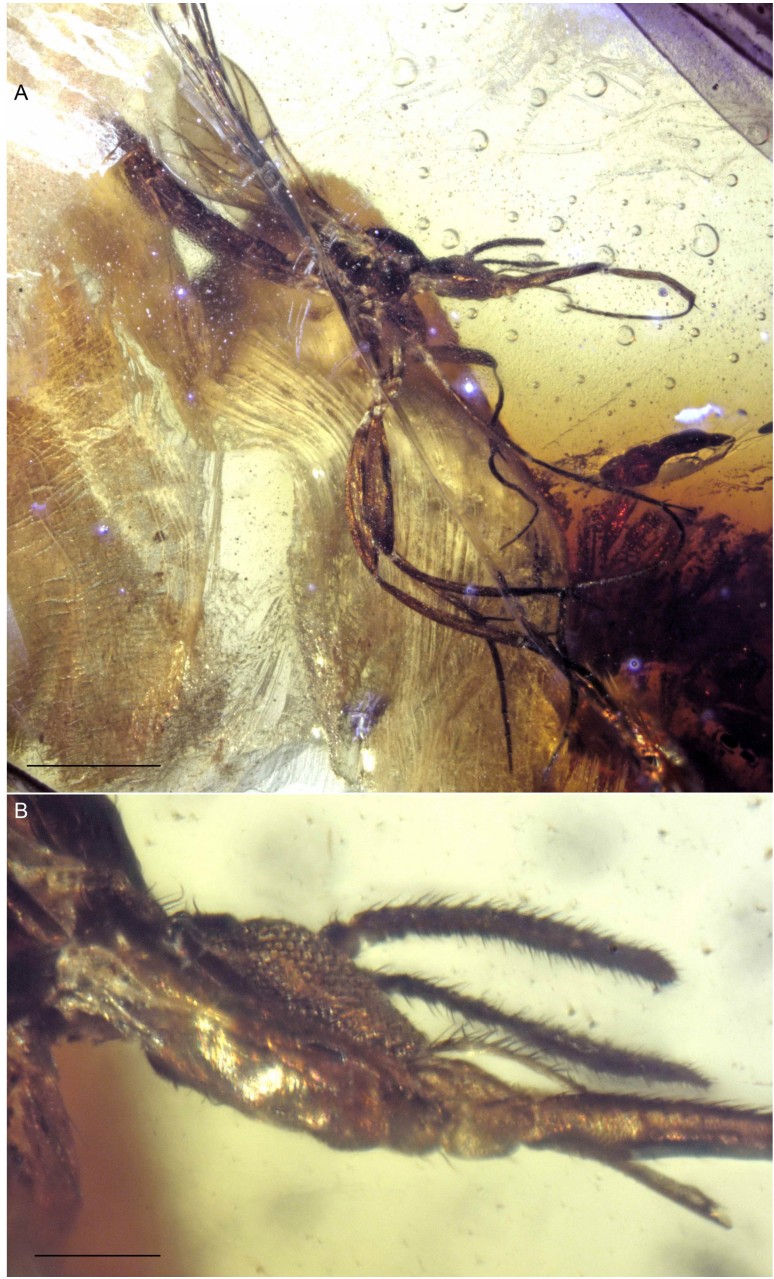

**Figure 4** Photographs of *Palaeognoriste orientale* n. sp., paratype female Tad-324. (A) Habitus. (B) Head. Scale bars: (A) 0.5 mm, (B) 0.1 mm.

of other spur. Hind tibia with apico-lateral comb of thick, short setae (presence/absence of depression here not observable). Hind coxa without lateral concavity. Tips of all claws pointed, none blunt.

**Wing** (Fig. 7B): Fairly short and broad, length/width 2.13, apex rounded but not extensively so. $R_1$ setose, $R_5$ without setae, other veins bare. Costa extending to approximately 2/3 distance between $R_5$ and $M_1$. Sc ending at C. $R_1$ slightly curved posteriorly, apex ending beyond middle of wing. $R_5$ moderately sinuous in apical third. M stem and bases of $M_{1+2}$

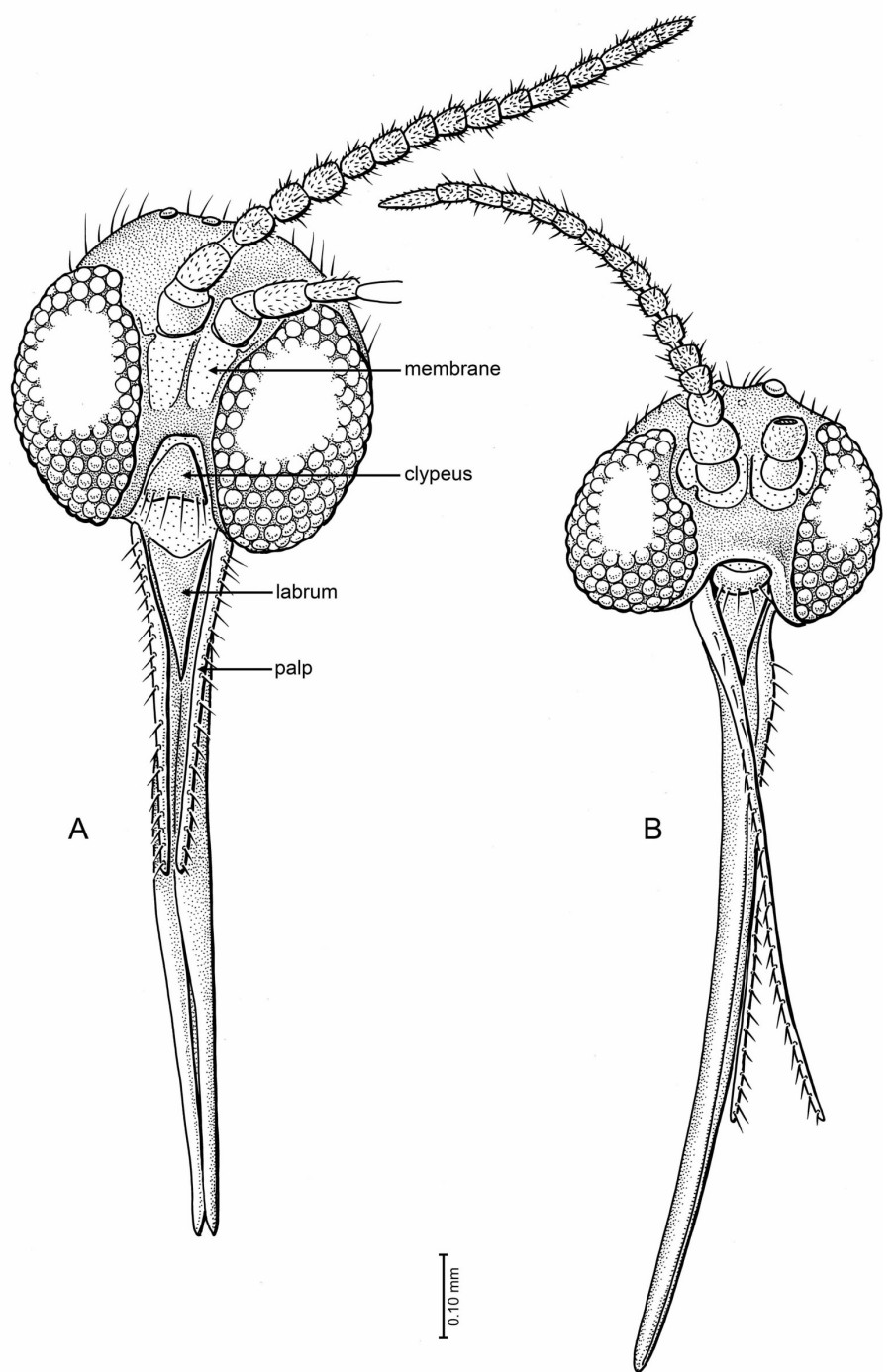

**Figure 5** **Drawings of frontal view of heads (same scale).** (A) *Indorrhina sahnii* n. sp., holotype Tad-418 (B) *Lygistorrhina indica* n. sp., holotype Tad-442.

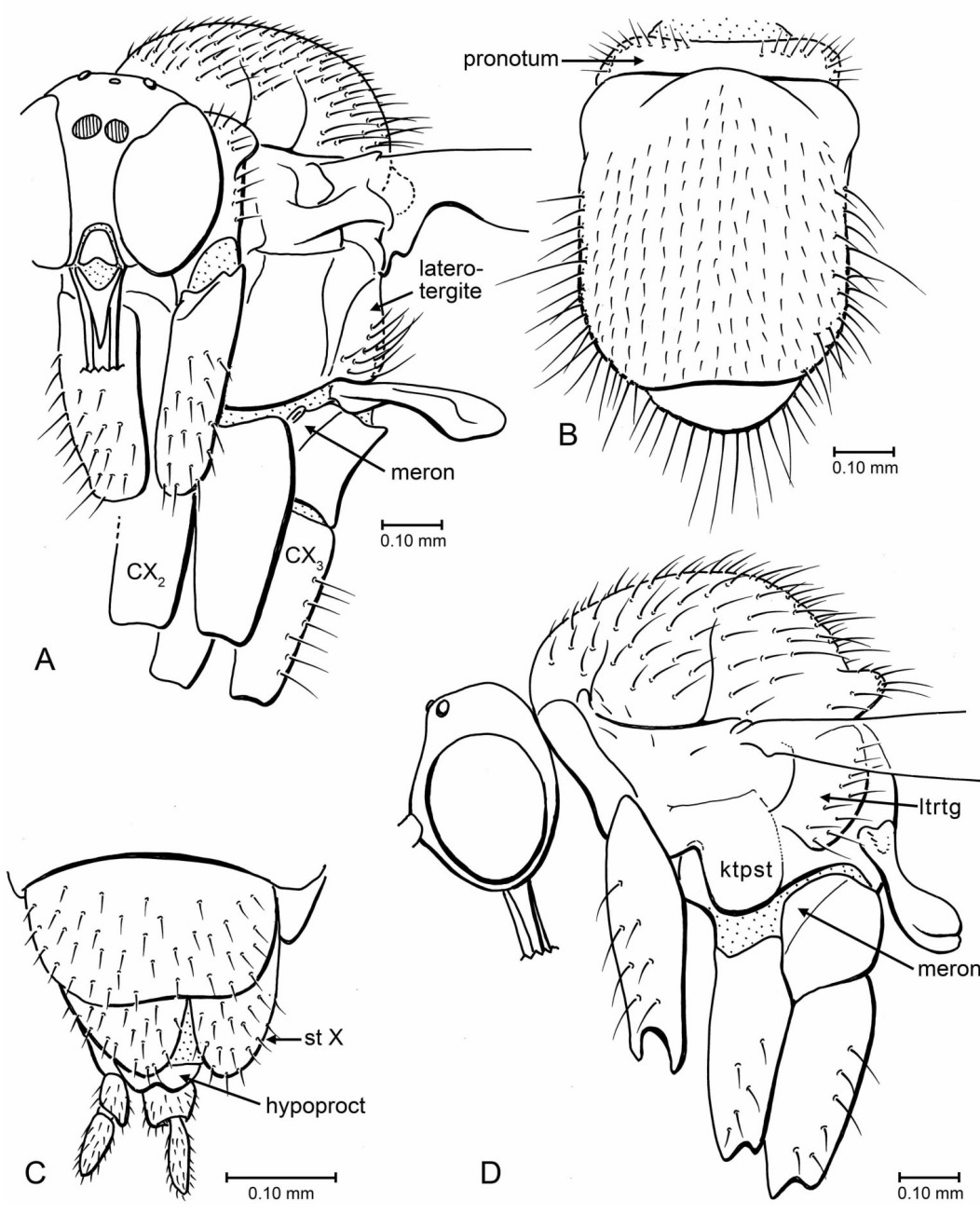

**Figure 6 Drawings of head, thorax and terminalia.** (A) *Indorrhina sahnii* n. sp., holotype Tad-418, head and thorax, fronto-lateral view. (B) *Indorrhina sahnii* n. sp., holotype Tad-418, thorax, dorsal. (C) *Lygistorrhina indica* n. sp., holotype Tad-442, female terminalia, ventral. (D) *Lygistorrhina indica* n. sp., paratype Tad-492, thorax and head, lateral (thorax reconstructed from both sides). Only the base of the proboscis is shown in (A, D).

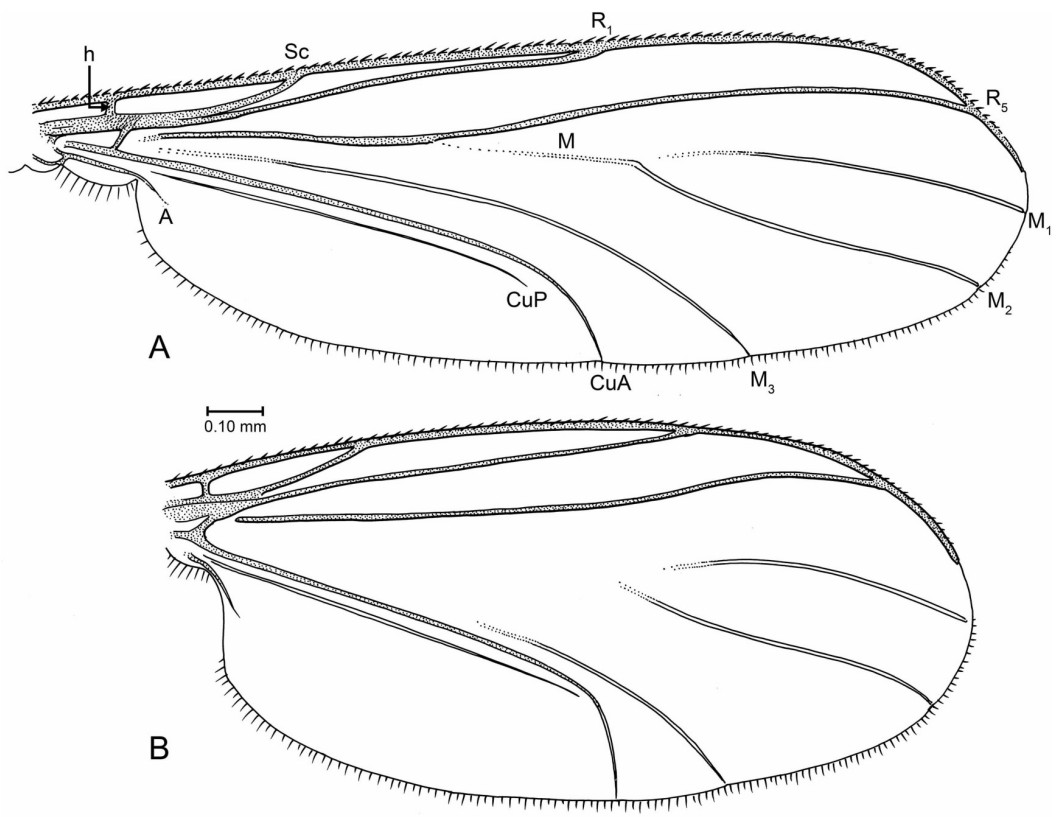

**Figure 7** **Drawings of wings (same scale).** (A) *Indorrhina sahnii* n. sp., holotype Tad-418. (B) *Lygistorrhina indica* n. sp., paratype Tad-492.

fork lost. $M_2$ extends slightly more basally than $M_1$. CuP very close and parallel to stem of CuA; apex of CuA strongly curved, almost perpendicular to stem of CuA. Anal lobe well developed.

**Abdomen**: That of female long, slender, tubular, cerci two-segmented (Fig. 6C; male unknown.

**Additional remarks:**

**Tad-442** (Figs. 5B, 6C, 7B, 8A, 8B): Holotype female; preservation: The specimen is very well preserved and virtually complete save for the distal tarsomeres of 4 legs (which are lost at the amber surface). Head and its appendages are very well preserved and fully observable; right wing is slightly distorted by preservation, but the left wing is well preserved. A fully lateral view (e.g., thorax) is not observable.

**Tad-492** (Figs. 6D, 8C, 8D): Paratype female; preservation: Specimen is largely complete but not intact; dorsum of the thorax and abdomen are separated from the rest of the body along with the wings (which are still attached to the thorax); a fracture along the line of separation prevents a full lateral view of the intact portion of the pleurites. Otherwise, the head and its appendages, legs, wings, and terminalia are well preserved with many details observable.

Measurements: Head height: 0.38 mm; proboscis length: 0.90 mm; palpus length: 0.45 mm; antenna length: 0.68 mm; body length without head: 2.64 mm; wing length from humeral vein: 1.83 mm.

**Tad-888** (Figs. 9A, 9B): Paratype, sex unknown; preservation: Most of abdomen and mid and hind legs missing; both wings incomplete.

Measurements: Head height: 0.22 mm; antenna: 0.5 mm; proboscis: 0.5 mm; palpus: 0.35 mm.

### *Indorrhina* Stebner and Grimaldi, new genus
LSID urn:lsid:zoobank.org:act:D8BCCDD4-D601-467A-9478-482E3ADF269C
Figs. 5A, 6A, 6B, 7A, 10A–10D

**Diagnosis**: Proboscis moderately long (ca. 2.17× head length); antenna with 14 flagellomeres, articles cylindrical in shape; antenna relatively short (1.8× length of head); median ocellus small, in nearly transverse line with lateral ones; laterotergite lobate, setose; wing relatively long and narrow, length/width 2.8; Sc complete; stem of M and bases of $M_2$ branch present but faint to spectral; hind coxa without lateral concavity.

**Type Species**: *Indorrhina sahnii*, n.sp.

**Comments**: The new genus appears to be phylogenetically intermediate between *Asiorrhina* and *Lygistorrhina*. The proboscis length is shorter than in *Lygistorrhina*, more similar to that of *Asiorrhina* (2.05–1.75× head depth). Unlike *Asiorrhina*, however, the antenna of *Indorrhina* is not laterally flattened and is considerably shorter (1.8× head depth, vs. 2.5× in *Asiorrhina*). Since the male of *Indorrhina* is unknown, it cannot be checked if the gonostylus is apically forked, a unique characteristic of *Asiorrhina*. Conditions of the ocelli and laterotergite in *Indorrhina* are shared with *Lygistorrhina*. *Indorrhina* differs from that genus by a hind tibia that is not apically broadened, lack of a concavity on the hind coxa, the shorter proboscis, and by retaining bases to M veins, a feature that also separates it from the two other Eocene taxa *Palaeognoriste* and *Parisognoriste*.

**Etymology**: Prefix based on India; suffix, -rrhina, from the Greek for nose or proboscis, in reference to the slender proboscis.

### *Indorrhina sahnii* Stebner and Grimaldi, new species
LSID urn:lsid:zoobank.org:act:4380AECC-6418-4EB0-BB1B-FC5E12CCCC8F
Figs. 5A, 6A, 6B, 7A, 10A–10D

**Diagnosis:** As for the genus, by monotypy.

**Material**: Holotype, female, Tad-418, in Early Eocene Cambay amber, deposited in the BSIP.

**Locality**: Cambay Formation (early Eocene), Tadkeshwar lignite mine, Tadkeshwar, Gujarat, India, 21°21.400′N, 73°4.532′E.

**Etymology**: Patronym for Professor Ashok Sahni, dean of Indian paleontology, who provided advice and encouragement to the authors for research on Cambay amber.

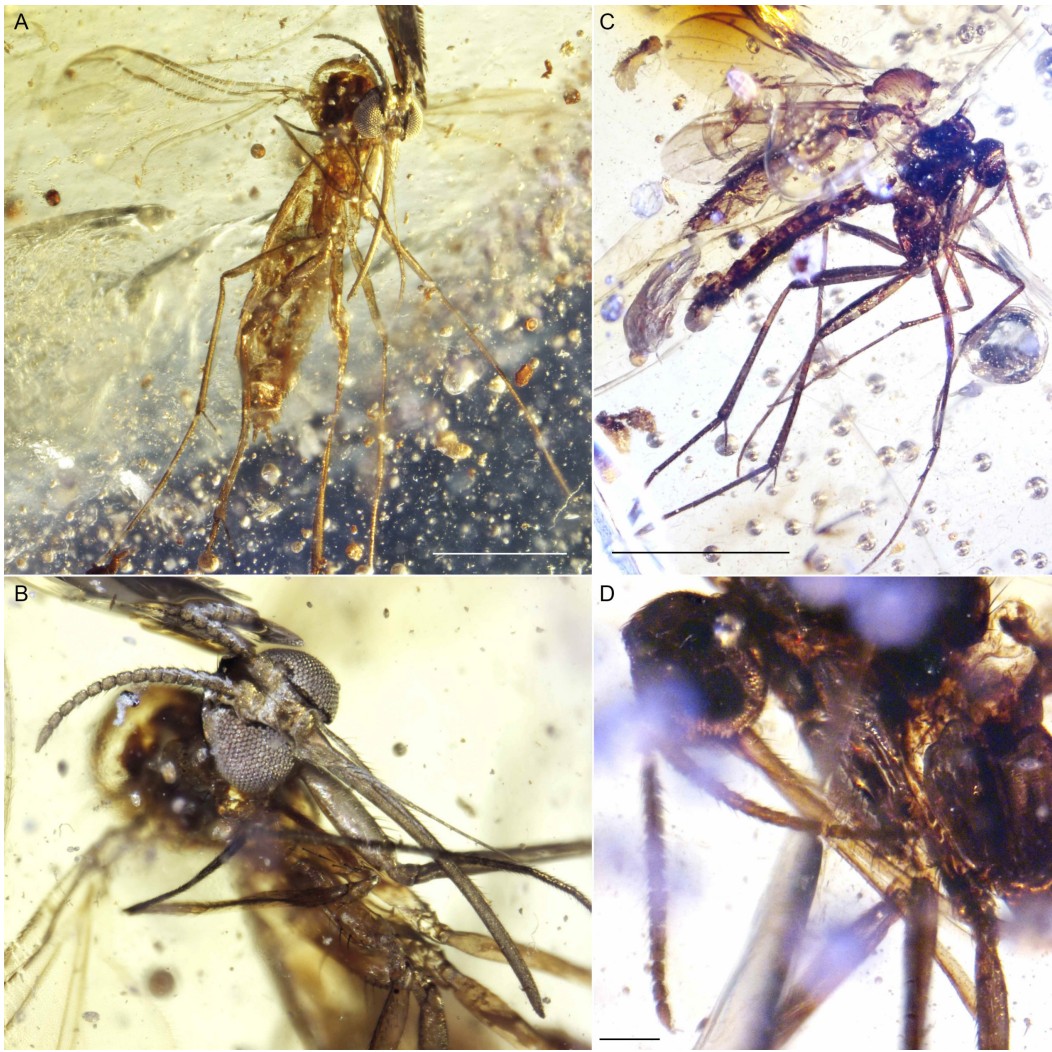

**Figure 8** **Photographs of _Lygistorrhina indica_ n. sp.** (A) _Lygistorrhina indica_ n. sp., holotype female Tad-442, habitus, fronto-lateral view. (B) _Lygistorrhina indica_ n. sp., holotype female Tad-442, head, frontal view. (C) _Lygistorrhina indica_ n. sp., paratype Tad-492, habitus, lateral view. (D) _Lygistorrhina indica_ n. sp., paratype Tad-492, head, lateral view. Scale bars: (A) 0.5 mm, (C) 1 mm, (D) 0.1 mm.

**DESCRIPTION**: Based on unique female, Tad-418 (holotype).

**Measurements**: Head height: 0.39 mm; body without head: 2.68 mm; wing length, from humeral vein: 1.94 mm; antenna length: 0.76 mm; proboscis length: 0.81 mm; palpus length: 0.43 mm.

**Head** (Fig. 5A): Slightly flattened antero-posteriad (not subspherical), occipital and postocellar setae well developed, frontal setae absent. Eyes well separated (frons well developed), with slight medial emargination near antennal base; no differentiation of facets; interocular setulae very short, sparse, and fine. All three ocelli present, median ocellus smaller than lateral ones, in nearly transverse line. Face well developed, with a large membranous area beneath each antennal base. Oral margin very deeply incised, with clypeus (roughly trapezoidal in shape) closely fitting into oral margin; clypeus flat, surrounded by

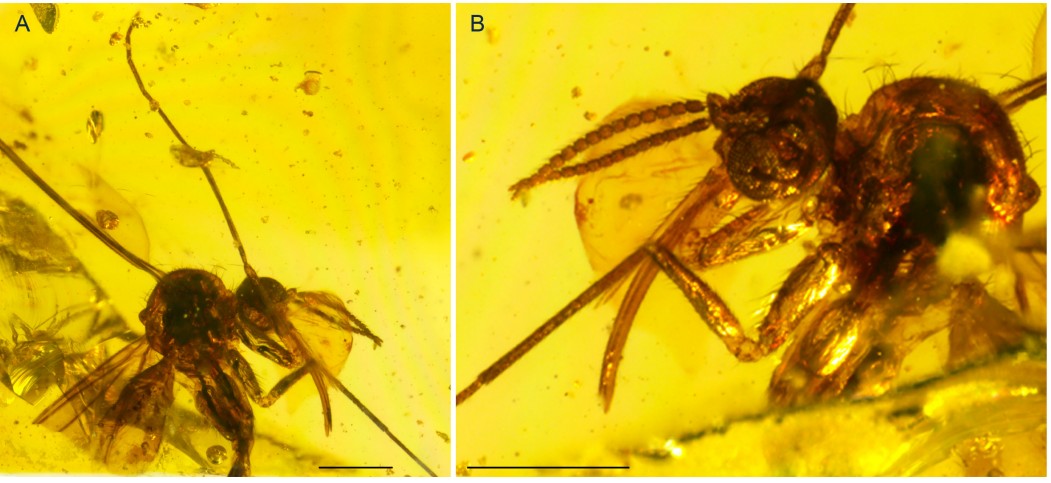

**Figure 9** Photographs of *Lygistorrhina indica* n. sp. (A) *Lygistorrhina indica* n. sp., paratype Tad-888, habitus, lateral view. (B) *Lygistorrhina indica* n. sp., paratype Tad-888, thorax and head. Sale bars: (A) 0.5 mm, (B) 0.5 mm.

membrane, anterior margin with row of 6 setae. Palp moderately long, length 0.5× that of proboscis; 1-segmented, with longitudinal row of >20 setae along all of lateral surface; palp slightly tapered apicad. Labrum long triangular, anterior margin concave, glabrous. Proboscis moderately long, length 2.17× the head depth, relatively stout (particularly at base), longer than fore femur. Antenna with 14 flagellomeres, total length relatively short (1.8× depth of head). Scape bare, barrel-shaped; pedicel longer than wide; flagellomeres longer than wide, lengths greater in apical four articles, cylindrical (not laterally flattened); each flagellomere with subapical whorl of setae; apical flagellomere tapered.

**Thorax** (Figs. 6A, 6B): Scutum convex but not strongly arched or dome-shaped; entirely covered with setae. Acrostichals long, arranged into fairly even rows; long setae on lateral margins of scutum. Scutellum short and broad, with row of 12–14 long setae on posterior margin, no setae on dorsal surface. Pleura mostly bare. Laterotergite lobate, with posterior and ventral margins raised above pleural wall; setose, setae not confined to posterior margin.

**Legs**: Fore coxa slightly longer than other coxae; fore coxa and hind coxa setose; mid coxa bare; hind coxa without lateral concavity. Tibiae and tarsi with longitudinal rows of setulae. Fore tibia slightly longer than fore femur. Tibial spurs 1 : 2 : 2. Hind tibia with short, transverse comb of thick setae at apex. All pretarsal claws with apices pointed, none blunt.

**Wing** (Fig. 7A): Wing relatively long and narrow, length/width 2.8, apex of wing not broadly rounded. Vein C extended to approximately 2/3 distance between $R_5$ and $M_1$. Sc complete; $R_1$ ending slightly beyond mid-length of wing. Presence/absence of setulae on R veins not observable. Stem of M and base of $M_2$ fork present but faint to spectral, base of $M_1$ lost; $M_1$ and $M_2$ parallel for most their length; $M_1$ ending at wing apex. $M_{3+4}$ largely intact, most of stem present (though incomplete); CuA long, apically curved but not acutely so; CuP close and parallel to CuA. Anal lobe present but not highly developed.

**Abdomen**: Long, slender, tubular; cerci 2-segmented.

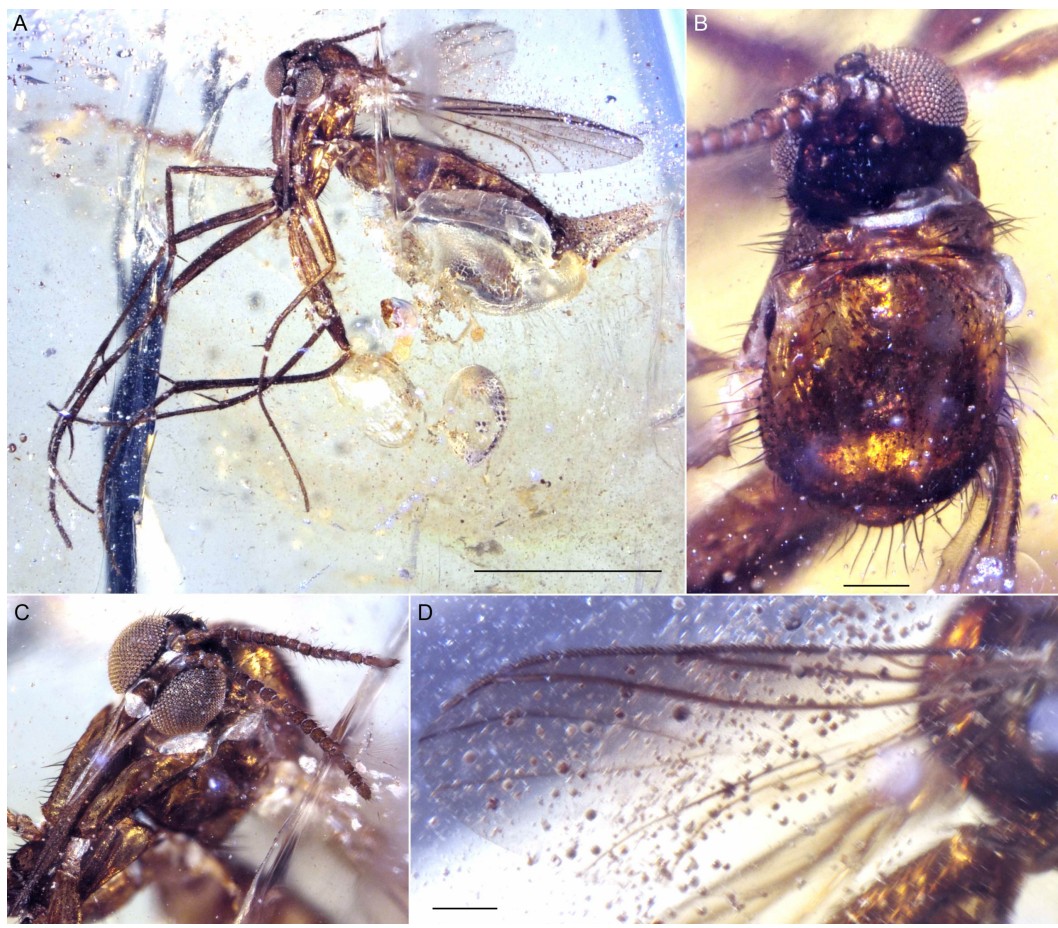

**Figure 10** **Photographs of *Indorrhina sahnii* n. gen et sp., holotype female Tad-418.** (A) habitus, lateral view. (B) Thorax and head, dorsal view. (C) Head, frontal view. (D) Wing. Scale bars: (A) 1 mm, (B) 0.1 mm, (D) 0.1 mm.

**Remarks:** Holotype Tad-418; preservation: Specimen is complete and fully intact, though portions are obscured: the ventral abdominal region by a bubble, and portions of the mid- and hind tibiae and tarsi from a fracture through the piece. The head, proboscis, thorax, and wings are particularly well preserved, with no collapse or crumpling of the cuticle. A full frontal view of the head is observable, but only an oblique view of the thoracic plura is possible.

## DISCUSSION

With eight fossils in three species and three different genera the diversity of Lygistorrhinidae in Cambay amber exceeds that in Baltic amber, which has revealed only two species within the genus *Palaeognoriste* so far (*P. affine* Meunier, 1912 and *P. sciariforme* Meunier, 1904). This is most surprising considering the profound differences in sampling. Baltic amber is the best studied amber deposit in the world. Being mined commercially for decades, probably tens-of-thousands of insect inclusions are sorted out every year. In contrast, Cambay

amber, which has been collected and studied only a few years, has revealed about 2,000–3,000 insect inclusions so far, overall based on work in four labs (BSIP, AMNH, Steinmann Institute Bonn, University of Kansas). A reasonable explanation for the greater diversity of Lygistorrhinidae in Cambay amber may be found in the climatic conditions prevailing at the time of amber formation. Recent Lygistorrhinidae are essentially circumtropical, occurring around the world in tropical regions (even remote islands), their northernmost extensions being into wet, warm temperate areas of southeast US (*Lygistorrhina sanctacatherinae*) and Japan (*L. pictipennis*). At the time of amber formation the Indian subcontinent was just entering the equatorial humid-belt (e.g., *Kent & Muttoni, 2008*), resulting in an extremely hot and humid tropical climate. Virtually all plants recorded from Vastan mine are restricted to tropical and subtropical regions today, i.e., they grow under very warm and very humid conditions. It has therefore been concluded that the palaeoflora is indicative of a warm and very humid climate, with certain elements such as dipterocarps suggesting a tropical rainforest growing in the vicinity of Vastan (*Singh et al., 2015*; *Tripathi & Srivastava, 2012*). More precisely, the reconstruction depicts a terrestrial lowland environment with a mesophytic mixed forest growing under tropical climate and with sufficient humidity (*Singh et al., 2015*). This general reconstruction of the Eocene palaeoforest in India fits well with the ecological requirements of extant representatives of Lygistorrhinidae.

The fossil record of Lygistorrhinidae shows a high congruence with the cladistic results, i.e., earlier fossils represent only older lineages (see phylogenetic analysis in *Blagoderov & Grimaldi, 2004*; *Blagoderov, Hippa & Nel, 2010*). The genera only known from Cretaceous ambers seem to belong to a basal stem group, whereas *Palaeognoriste*, which is only known from the Eocene, presumably represents a sister group to all the extant genera (*Blagoderov & Grimaldi, 2004*; *Blagoderov, Hippa & Nel, 2010*). The taxa found in Cambay amber clearly support this by the absence of 'Cretaceous' taxa and by the presence of the 'Eocene' taxon *Palaeognoriste*, as well as the presence of the phylogenetically younger extant genus *Lygistorrhina*, and the fossil genus *Indorrhina* n. gen., which appears to be within the clade of extant taxa that form the sister group to *Palaeognoriste* (Fig. 11).

*Paleognoriste orientale* n. sp. possesses at least 7 features that differ with the two species of the genus in Baltic amber: broader, shorter wing; stem of M and base of M forks absent; apex of vein CuA more acutely bent; gonostylus more curved; apex of labellar lobes blunt; clypeus projecting forward, Ω-shaped; antenna much longer than proboscis. However, given the overall similarity of the three species in proportions of body regions, appendages, and wing venation, these differences do not warrant erecting a separate genus. The finding adds another record to the pattern of apparently shared fossil taxa between Cambay and Baltic amber, as already been reported from the Diptera (*Grimaldi & Singh, 2012*; *Stebner et al., 2017*), bees (*Engel et al., 2013*), and termites (*Engel et al., 2011*).

The most unexpected finding is *Lygistorrhina* in Cambay amber. This is a clearly defined, unquestionably monophyletic genus. The only difference between the 21 Recent species and the Cambay species is that the latter lacks the shallow depression on the lateral surface of the metacoxa, and the setae on the laterotergite are not in a row on the posterior margin. Otherwise, *L. indica* n. sp. is clearly closely related to recent *Lygistorrhina*. This approximately triples the age of *Lygistorrhina*, the prior oldest record being a species

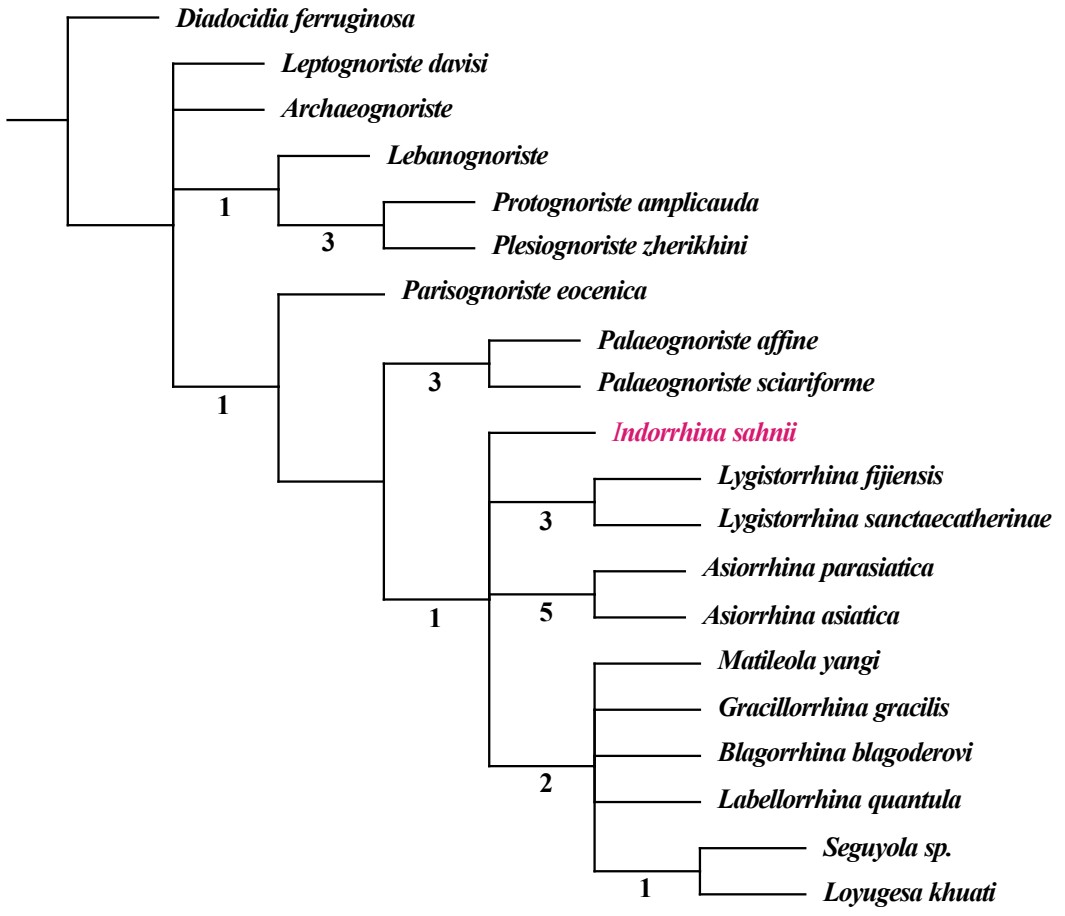

**Figure 11 Phylogeny of Lygistorrhinidae. Strict consensus cladogram of 9 MPT (181 steps), based on the matrix of *Blagoderov, Hippa & Nel (2010)* and including *Indorrhina sahnii* from Cambay amber.** Numbers below branches refer to Bremer support values.

in Miocene Dominican amber (*Grund, 2012*; there are at least four other species of *Lygistorrhina* in this amber (D Grimaldi, 2017, unpublished data)). Biogeographically, this finding is difficult to interpret because of the poor geographical sampling of the Recent species of *Lygistorrhina*. For example, there are nine New World species (but with nearly 30 total, based on study of museum specimens by D.G.); five species in central Africa and the Comoro Islands; three in southeast Asia; and three species in the Australasian region (including New Caledonia and the Fiji archipelago). There are few or no described species from large areas: most of Central America, all of the Greater Antilles, Andean South America, Madagascar, India, southern China, much of southeast Asia, most of Indonesia, and all of New Guinea. Based on study of extant material by the senior author (DG) there are likely to be well over 100 living species in existence. Unfortunately, living species are morphologically challenging to separate, and so morphology will probably yield an insufficient number of characters for an eventual phylogenetic study of the genus.

## CONCLUSIONS

Considering the scarcity of Eocene amber deposits, Cambay amber is of great significance because it fills a gap in the spatial fossil record of the Paleogene and provides information about phylogenetic relationships, divergence estimations, and biogeographic patterns of certain groups, and adds information about the palaeoenvironment.

The discovery of a diverse assemblage of Lygistorrhinidae in Cambay amber reinforces the reconstruction of a tropical palaeoenvironment and, at the same time, indicates climatic differences between the Cambay amber forest (tropical) and the Baltic amber forest, which was located distinctly further north and presumably after the peak of Eocene global warming.

The finding of *Palaeognoriste* in Cambay amber adds further evidence that faunal exchange between India and Europe/Asia occurred before the formation of Cambay amber in the early Eocene. In this regard, the discovery of Lygistorrhinidae in Eocene Fushun amber from northeast China, Paleocene amber from Sakhalin Island, Russia, Rovno amber from the Ukraine, and any additional species in the Eocene amber from Oise, France would be very interesting for further studies.

The oldest finding of *Lygistorrhina* clearly has major implications for estimates of divergence times in the family as it indicates that this group was in existence since at least the early Eocene, which may explain its worldwide distribution, including some remote islands.

## ACKNOWLEDGEMENTS

The authors thank Ashok Sahni (Centre of Advanced Study in Geology, Panjab University, Chandigarh, India) and Rajendra. S Rana (Department of Geology, Hemwati Nandan Bahuguna Garhwal University, Srinagar, India) for support in field work. The authors are grateful to the authorities of the Tadkeshwar lignite mine for assistance during fieldwork in Gujarat, India. HS thanks the director of the Birbal Sahni Institute of Palaeosciences (Lucknow, India) for his support. FS thanks Simon Gunkel (Steinmann Institut, University of Bonn, Germany) for professional support.

### Funding

The research was made possible with funding provided for Jes Rust (Steinmann Institut, Universität Bonn, Germany) No. RU665/10-1 from the German Research Foundation (DFG). Hukam Singh is supported by the Department of Science and Technology New Delhi (DST) (Project No. EEQ/2016/000112). The funders had no role in study design, data collection and analysis, decision to publish, or preparation of the manuscript.

### Grant Disclosures

The following grant information was disclosed by the authors:
Jes Rust (Steinmann Institut,Universität Bonn, Germany): RU665/10-1.
Department of Science and Technology NewDelhi (DST): EEQ/2016/000112.
## Competing Interests

The authors declare there are no competing interests.

## Author Contributions

- Frauke Stebner analyzed the data, contributed reagents/materials/analysis tools, wrote the paper, prepared figures and/or tables, reviewed drafts of the paper.
- Hukam Singh contributed reagents/materials/analysis tools, wrote the paper.
- Jes Rust analyzed the data, contributed reagents/materials/analysis tools, wrote the paper, reviewed drafts of the paper.
- David A. Grimaldi analyzed the data, contributed reagents/materials/analysis tools, wrote the paper, prepared figures and/or tables, reviewed drafts of the paper.

## Data Availability

The raw data is included in the manuscript.

## New Species Registration

The following information was supplied regarding the registration of a newly described species:

Publication LSID:

urn:lsid:zoobank.org:pub:DD2F5D2A-C3EE-4F5C-B068-C9E541D9B70C

Palaeognoriste orientale Stebner and Grimaldi, new species

LSID urn:lsid:zoobank.org:act:122BEB14-EDDA-45D7-9973-3898B609F39C

Lygistorrhina indica Stebner and Grimaldi, new species

LSID urn:lsid:zoobank.org:act:C787AB55-78E4-4E19-940D-29BEAD5A2ABE

Indorrhina Stebner and Grimaldi, new genus

LSID urn:lsid:zoobank.org:act:D8BCCDD4-D601-467A-9478-482E3ADF269C

Indorrhina sahnii Stebner and Grimaldi, new species

LSID urn:lsid:zoobank.org:act:4380AECC-6418-4EB0-BB1B-FC5E12CCCC8F.

## Supplemental Information

Supplemental information for this article can be found online at http://dx.doi.org/10.7717/peerj.3313#supplemental-information.

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
