# Peer review of "Lygistorrhinidae (Diptera: Bibionomorpha: Sciaroidea) in early Eocene Cambay amber"

_PeerJ, doi:10.7717/peerj.3313_

## Round 0.1 · original submission · Minor Revisions

As demonstrated by the assessments by the reviewers and myself, the manuscript is in a good state. There are, however, still some minor points I would like you to address, which could make the manuscript even better:

• Phylogeny: I agree with reviewer 1 that it might be useful to add the genus to the recent phylogeny by Blagoderov et al. 2009 (considering most characters have already been coded, it should not take that much additional effort to add the additional genus and test it placement in the tree). Reviewer three also pointed out similarities (R-M fusion) to some genera of Keroplatidae, which might be worth discussing.

• Listed amber age: As far as I am aware, ages of many amber deposits are not entirely uncontroversial (Dominican amber has by some been dated up to 40 Ma, although it is now mostly accepted that it was deposited between 20 – 15 Ma). I would therefore be useful to have a graph illustrating the ages of all discussed ambers as these are directly relevant for your interpretations (climate and biogeography). It would be at least necessary to cite the works which list the ages used in the text (Oise Amber, Baltic Amber, etc.) and what those ages are based on.

In addition to the (additional) points raised by the reviewers, please also address the following points:

Line 39: it might be good to point that Seguyola is derived in this context for completeness sake
Line 58-60: please provide reference for the ages you list for these amber deposits. It think the Oise Amber is not that precisely dated, so add “ca.”. As you give absolute age for the rest of the deposits, it would be useful to give an absolute age range for the Baltic amber.
Line 78: Unclear, what you mean with “diverse in past and present ecosystems than is commonly known”. How can these fossils tell you anything about the present? Do you mean that “these flies might have much more abundant in past ecosystems than today”.
Line 89-91: One can also wonder if the really are off the same age; please be more specific about this issue.
Line 140: I find it a bit confusing to talk about other species in the Diagnosis, but maybe this is more common in insects. In mollusks, one would typically compare taxa in the Discussion.
Line 406-408: Are there no other (additional) references for the reconstration as a tropical rain forest.
Line 418-419: It might be useful to add the new genus in the phylogeny provided by Blagoderov et al. (2009).
Line 434: It is a bit weird to speak about sister species if you did not actually test this.

·

Basic reporting

clear and unambiguous
references quite ok
structure quite good
relevant results and well supported conclusions

a very good paper, worth being published in the state

Experimental design

no comment

Validity of the findings

no comments, results robust

maybe, a point could improve the paper, that is including the new genus in the available phylogeny
but this is only optional

Additional comments

a very good paper, without errors

maybe, a point could improve the paper, that is including the new genus in the available phylogeny
but this is only optional

Reviewer 2 ·

Basic reporting

no comment

Experimental design

no comment

Validity of the findings

No comment

Additional comments

Is an excellent manuscript with very adequate descriptions and illustrations that contribute significantly to the knowledge of the family Lygistorrhinidae.

I add some comments directly to the text.

Annotated reviews are not available for download in order to protect the identity of reviewers who chose to remain anonymous.

·

Basic reporting

It is a well written manuscript by the experienced authors. I have not found any errors, except in the References where are several typos in the names of the journals (e.g. in Hennig 1966, Matile 1979, Matile 1990b, Matile 1996). All the literature is relevant and properly cited. The figures are of high quality, with appropriate legends.

Experimental design

The authors followed standard procedures of systematic paleoentomology. The fossil specimens are preserved in publicly accessible collections.
My only concern is the description of new taxa based just on females in the groups like Sciaroidea where male terminalia provide important diagnostic and phylogenetic characters. In extant species, it is usually better to leave the species unnamed, as was done e.g. by Matile (1990a) for Lygistorrhina from Madagascar. Possibly in the case of fossils, such a practice can be tolerated, considering the rarity of well-preserved specimens and low probability that a closely related species from the same genus will be found in the future.

Validity of the findings

The new taxa appear to be valid and they are well supported by their diagnostic characters. The results are properly discussed and most of the new findings are shown to be of high phylogenetic and zoogeographical significance.

Additional comments

Lygistorrhinids belong to the least known groups of fungus gnats (Sciaroidea), being one of the few extant families of Diptera where immature stages are completely unknown and where the phylogenetic position is still more or less obscure. Although beyond the scope of this manuscript, it would be interesting to see the authors’ opinion about the phylogenetic position of this peculiar family within Sciaroidea, with respect to fossil taxa. The older the fossils of Lygistorrhinidae are, the more difficult is their delimitation against the other Sciaroidea, especially regarding the family Pleciomimidae (= Antefungivoridae) and the unplaced (incertae sedis) genera of Sciaroidea. Some of the genera from the Cretaceous ambers (see Blagoderov & Grimaldi 2004), referred to as belonging to Lygistorrhinidae, have apparently little in common with recent lygistorrhinids and could be well placed in the family Mycetophilidae or possibly to other groups.
The new genus Indorrhina deserves special attention as it retains the stem of M-fork and its probable fusion with R-stem (as far as this can be seen on the holotype). This feature separates it from both Palaeognoriste and Parisognoriste and should perhaps be more discussed in the Comments paragraph of the description (page 14 of the reviewing PDF). The possible R-M fusion would indicate its relationship to some genera of Keroplatidae, especially to Microkeroplatus, Chetoneura and Pseudochetoneura.

---

## Round 0.2 · accepted · Accept

Thank you for addressing our final suggestions. The requirements for describing new species are met.